# Re-expression of SynGAP protein in adulthood improves translatable measures of brain function and behavior

Thomas K Creson[1,2†], Camilo Rojas[1,2†], Ernie Hwaun[3], Thomas Vaissiere[1,2], Murat Kilinc[1,2], Andres Jimenez-Gomez[4,5], Jimmy Lloyd Holder Jr[4,5], Jianrong Tang[4,5], Laura L Colgin[3], Courtney A Miller[1,2], Gavin Rumbaugh[1,2]*

[1]Department of Neuroscience, The Scripps Research Institute, Jupiter, United States; [2]Department of Molecular Medicine, The Scripps Research Institute, Jupiter, United States; [3]Department of Neuroscience, Institute for Neuroscience, Center for Learning and Memory, University of Texas at Austin, Austin, United States; [4]Jan and Dan Duncan Neurological Research Institute, Baylor College of Medicine, Houston, United States; [5]Department of Pediatrics, Baylor College of Medicine, Houston, United States

*For correspondence:
grumbaug@scripps.edu

[†]These authors contributed equally to this work

**Abstract** It remains unclear to what extent neurodevelopmental disorder (NDD) risk genes retain functions into adulthood and how they may influence disease phenotypes. *SYNGAP1* haploinsufficiency causes a severe NDD defined by autistic traits, cognitive impairment, and epilepsy. To determine if this gene retains therapeutically-relevant biological functions into adulthood, we performed a gene restoration technique in a mouse model for *SYNGAP1* haploinsufficiency. Adult restoration of SynGAP protein improved behavioral and electrophysiological measures of memory and seizure. This included the elimination of interictal events that worsened during sleep. These events may be a biomarker for generalized cortical dysfunction in *SYNGAP1* disorders because they also worsened during sleep in the human patient population. We conclude that SynGAP protein retains biological functions throughout adulthood and that non-developmental functions may contribute to disease phenotypes. Thus, treatments that target debilitating aspects of severe NDDs, such as medically-refractory seizures and cognitive impairment, may be effective in adult patients.
DOI: https://doi.org/10.7554/eLife.46752.001

## Introduction

Neurodevelopmental disorders (NDDs), including intellectual disability (ID), autism spectrum disorder (ASD), and epilepsy result in treatment-resistant behavioral abnormalities. Many severe NDDs are characterized by cognitive impairments and medically-refractory seizures (*Boyle et al., 2011*; *Jeste and Tuchman, 2015a*). In some cases, seizures, and underlying circuit-level excitatory imbalances that trigger these events, are thought to contribute to worsening of cognitive and behavioral phenotypes (*Scheffer et al., 2017*). Therefore, it is crucial to develop treatment strategies that improve the function of neural circuits associated with seizure susceptibility and cognitive impairment in NDD patient populations.

Historically, the patho-neurobiology underlying NDD symptomatology is thought to arise from impaired brain development. However, studies in animal models of genetic risk factors causally-linked to syndromic NDDs have shown improvement in meaningful measures of brain function and behavior in response to adult-initiated therapeutic interventions (*Guy et al., 2007*; *Ehninger et al., 2008a*; *Cui et al., 2008*). These findings suggest that not all phenotypic consequences of NDDs arise

through impaired neurodevelopment indicating that therapeutic intervention may be beneficial in adult NDD patients with fully mature brains (*Zoghbi and Bear, 2012*; *Ehninger et al., 2008b*). The potential impact of adult initiated treatments in NDD patients cannot be overstated, as there are currently no effective treatments for the most debilitating aspects of these disorders. Thus, hope remains that adults with NDDs could benefit from emerging therapeutic strategies.

Adult reversal studies have been carried out in animal models for only a few syndromic NDD genes (*Ehninger et al., 2008b*). In these models, with some notable exceptions, phenotypic reversal in adulthood is most effective in correcting social behaviors and/or motor function (*Guy et al., 2007*; *Ehninger et al., 2008b*; *Marín, 2016*; *Mei et al., 2016*). However, there is less evidence that adult-initiated treatments can improve both cognitive dysfunction and seizure susceptibility in animal models for human disorders that are defined by these core phenotypes. Over the past decade, hundreds of new genes have been linked to NDDs (*Vorstman et al., 2017*). Large-scale exome sequencing projects in children with classically undefined and sporadic NDDs have identified a pool of genes that infer 100% (i.e. causal) risk for developing a severe disorder caused by brain dysfunction (*Deciphering Developmental Disorders Study, 2015*; *Deciphering Developmental Disorders Study, 2017*; *Kyle Satterstrom et al., 2018*). Some of these newer NDD genes account for a significant fraction of total cases, and dramatic phenotypes, such as severe cognitive impairment and medically-refractory seizures, define these single gene disorders (*Deciphering Developmental Disorders Study, 2015*; *Deciphering Developmental Disorders Study, 2017*). Given that the list of completely penetrant NDD genes has expanded considerably over the last decade, it is critical to understand the effectiveness of adult-initiated treatments in animal models for these newly-discovered single gene disorders. Moreover, it is important to gauge to what extent emerging treatment strategies improve cognitive impairment and seizure susceptibility because these are two of the most debilitating outcomes associated with the most severe genetically-defined NDDs.

*SYNGAP1* is a recently discovered NDD gene (*Hoischen et al., 2014*; *Zhu et al., 2014*; *Hamdan et al., 2009*), causally-linked to a range of sporadic disorders, including ID (*Deciphering Developmental Disorders Study, 2015*; *Deciphering Developmental Disorders Study, 2017*; *Hamdan et al., 2009*; *Rauch et al., 2012*), ASD (*Kyle Satterstrom et al., 2018*; *O'Roak et al., 2014*; *Hamdan et al., 2011*), severe epilepsy (*Vlaskamp et al., 2019*; *Carvill et al., 2013*; *von Stülpnagel et al., 2015*) and schizophrenia (*Purcell et al., 2014*). De novo nonsense variants in *SYNGAP1* resulting in haploinsufficiency lead to a relatively frequent genetically-defined form of ID with epilepsy (termed MRD5; OMIM#603384). It has a reported incidence of 1-4/10,000 individuals, or 0.5–1.0% of ID cases (*Deciphering Developmental Disorders Study, 2015*; *Deciphering Developmental Disorders Study, 2017*; *Kyle Satterstrom et al., 2018*; *Berryer et al., 2013*; *Parker et al., 2015*), which is similar to the frequency of Fragile X syndrome. MRD5 patients express moderate-to-severe intellectual disability (IQ < 50), have severely delayed language development, and express some form of epilepsy and/or abnormal brain activity, with these manifestations appearing first in early childhood (*Vlaskamp et al., 2019*; *Berryer et al., 2013*; *Parker et al., 2015*; *Mignot et al., 2016*). *SYNGAP1* has been recognized as a high-priority risk gene worthy of in-depth study. This designation was first suggested based on its causal linkage to a broad range of neuropsychiatric disorders (*Hoischen et al., 2014*; *Zhu et al., 2014*). This notion is strengthened by the known biological functions of SynGAP protein. A major function of the protein is to integrate signaling through NMDA receptors with structural and functional synapse plasticity (*Kilinc et al., 2018*), which is a substrate shared among nearly all neuropsychiatric disorders (*Penzes et al., 2011*). Therefore, biological discoveries made in *Syngap1* mouse models may be broadly generalizable to idiopathic neuropsychiatric disorders.

*Syngap1* heterozygous knockout mice (Hets), which have ~50% reduction in SynGAP protein levels (*Clement et al., 2012*), offer both construct and face validity for MRD5 (*Kilinc et al., 2018*). *Syngap1* heterozygosity disrupts a developmental critical period, in which a normal ratio of excitatory and inhibitory synapses are formed in particular brain regions. Without this normal ratio, circuits can become overly excitable, which may contribute to pathological brain function (*Clement et al., 2012*; *Aceti et al., 2015*; *Clement et al., 2013*; *Ozkan et al., 2014*). Prior studies have shown that reduced *Syngap1* expression during this developmental critical period prevents the normal emergence of certain behavioral responses and cognitive functions because altered measures of working memory, locomotor activity, and anxiety are sensitive to neonatal reversal of *Syngap1* pathogenicity (*Aceti et al., 2015*) but resistant to similar approaches performed in adulthood (*Clement et al.,*

2012). However, knocking out SynGAP protein in the adult hippocampus causes memory impairments (*Muhia et al., 2012*) and increases seizure susceptibility (*Ozkan et al., 2014*), suggesting that *Syngap1* may also have unique, non-developmental functions in the adult brain that promotes cognitive function and suppresses neural excitability. *Syngap1* mice have an extensive endophenotype (*Clement et al., 2012*; *Ozkan et al., 2014*; *Guo et al., 2009*; *Muhia et al., 2010*). However, only a small subset of individual phenotypes have been tested for sensitivity to adult reversal of genetic pathogenicity in this line of mice (*Clement et al., 2012*). Based on these past results, we determined if robust disease-associated phenotypes in *Syngap1* mice, such as long-term memory and seizure susceptibility, are sensitive to a method of adult gene restoration that boosts pathologically low levels of SynGAP protein. Moreover, we also searched for neurophysiological correlates of behavioral alterations in *Syngap1* animals to gain insight into how protein re-expression in mature animals may improve brain function related to cognitive ability and seizure.

## Results

Epileptic encephalopathy (EE) is one outcome of pathogenic *SYNGAP1* variation (*Carvill et al., 2013*). EE is a childhood neurological condition where epileptiform activity is believed to cause progressive worsening of brain function. Consistent with an EE designation, a substantial portion of MRD5 patients exhibit regressive phenotypes and are medically refractory to anti-epileptic drugs (*Vlaskamp et al., 2019*). *Syngap1* mice also express electrographic and behavioral seizures (*Mignot et al., 2016*; *Ozkan et al., 2014*; *Weldon et al., 2018*). Therefore, these animals can be used as a model for understanding how seizure-related phenotypes associated with MRD5 are modified by adult-initiated therapies. To do this, we used a *Syngap1* heterozygous (Het) animal model that enables whole-body SynGAP protein re-expression at different stages of life (*Clement et al., 2012*; *Ozkan et al., 2014*). This model is created by crossing *Syngap1* Lox-stop Het mice (*Syngap1$^{+/ls}$*) to hemizygous mice expressing a tamoxifen (TMX)-inducible form of Cre recombinase. Upon Cre activation, an artificial exon containing a premature stop codon is excised from the *Syngap1* gene, leading to re-activation of the targeted allele and restoration of SynGAP protein levels. To test for seizure susceptibility, we used the flurothyl seizure induction paradigm, which has been used previously to uncover phenotypes in constitutive *Syngap1* Het knockout mice (*Clement et al., 2012*). In this paradigm, flurothyl exposure leads to gradually progressing seizure types, and seizure susceptibility is quantified by recording the latency to reach each distinct seizure stage. In TMX(-) mice, there was a main effect of *Syngap1* genotype (i.e. the presence of the Lox-Stop allele), but not with the presence of Cre-ER, in the first two stages of seizure (*Figure 1A*; *Figure 1—source data 1*). There was no interaction detected between these two factors during any of the stages. The pairwise comparisons indicated that both Cre(-) and Cre(+) *Syngap1* Lox-Stop Hets reached the first two stages of seizure significantly faster than the corresponding *Syngap1* WT mice (*Figure 1A*; *Supplementary file 1*), which is indicative of a reduced seizure threshold and replicates our previous findings in adult constitutive *Syngap1* Het knockout mice (*Ozkan et al., 2014*). Importantly, there was no statistical difference between Cre(-) and Cre(+) *Syngap1* Lox-Stop Hets, indicating that Cre remained largely inactive in Cre(+)TMX(-) mice (*Figure 1A*; *Supplementary file 1*).

In TMX-treated mice, for the first two seizure stages, there was again an effect of the Lox-Stop allele, but not for Cre-ER (*Figure 1B*). However, we detected an interaction between Lox-Stop and Cre-ER in this study (*Figure 1B*). This suggested that restoration of SynGAP protein in adulthood influenced seizure threshold measurements. Pairwise comparisons of all four genotypes demonstrated that TMX improved the seizure threshold in Cre(+) Lox-Stop mice. For example, Cre(+) *Syngap1* Lox-Stop Het mice, or mice with *Syngap1* pathogenicity reversed starting at PND60 (*Figure 1—figure supplement 1*; *Supplementary file 2*; *Figure 1—figure supplement 1—source data 1*), were not statistically different from Cre(+) WT *Syngap1* mice in the second seizure stage (*Figure 1B*; *Supplementary file 1*; *Figure 1—source data 2*). This demonstrates that TMX increased the latency to tonic-clonic seizure induction in Lox-Stop mice to WT levels. Consistent with this, the same comparison during the first seizure stage uncovered a large difference in effect size between Cre(+) WT and Cre(+) Lox-Stop mice. Furthermore, the latency to seizure induction in Cre(+) *Syngap1* Lox-Stop mice was significantly increased compared to Cre(-) *Syngap1* Hets in the first two stages of the test (*Figure 1B*; *Supplementary file 1*; *Figure 1—source data 2*). Together, these

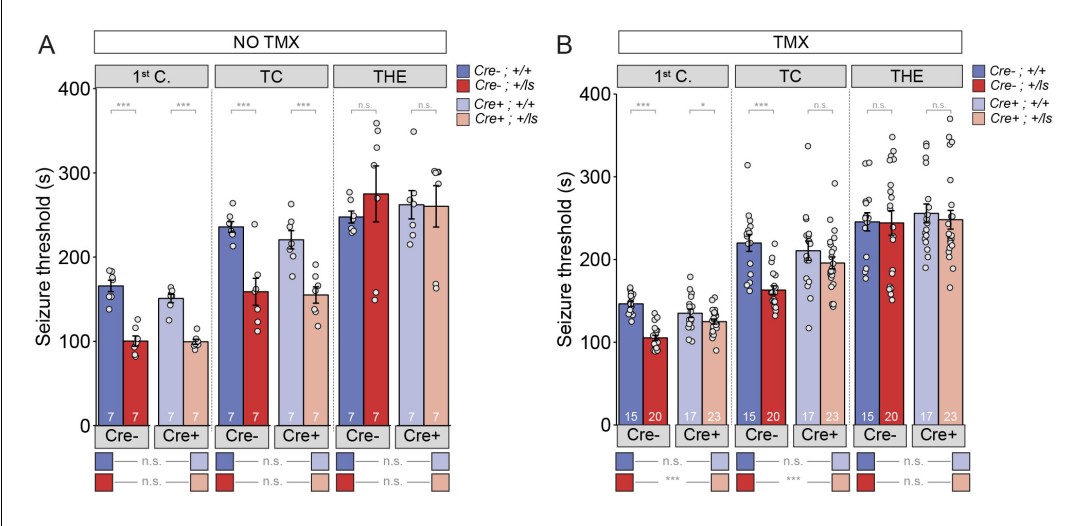

**Figure 1.** Seizure threshold is improved after adult restoration of SynGAP expression. (**A**) $Syngap^{Cre-;+/ls}$ and $Syngap^{Cre+;+/ls}$ mice exhibit hyperexcitability in two of the three events without Cre activation (No TMX) Main effects-1st clonus: Cre $F(1,24)=2.13$, $p=0.157$, Genotype $F = 117.73$, $p=9.75E-11$, Interaction $F(1,24)=1.69$, $p=0.206$); Cre- Cohen's $d = 3.855$, Cre +Cohen's $d = 4.737$. TC: Cre $F(1,24)=722$, $p=0.404$, Genotype $F(1,24)=40.05$, $p=1.53E-6$), Interaction $F(1,24)=.257$, $p=0.617$); Cre- Cohen's $d = 2.396$, Cre+ Cohen's $d = 2.405$. THE: Cre $F(1,24)=9.99E-6$, $p=0.998$, Genotype $F(1,24)=.320$, $p=0.577$), Interaction $F(1,24)=.420$, $p=0.523$. (**B**) $Syngap^{Cre+;+/ls}$ mice exhibit thresholds comparable to those of $Syngap^{Cre-;+/ls}$ mice after Cre activation (TMX-treated) in two of the three events Main effects-1st clonus: Cre $F(1,71)=2.59$, $p=0.112$; Genotype $F(1,71)=58.328$, $p=7.86E-11$, Interaction $F = 1 (1,71)=18.84$ $p=4.62E-5$; Cre- Cohen's $d = 3.329$, Cre+ Cohen's $d = 0.674$; TC: Cre $F(1,71)=4.53$, $p=0.037$, Genotype $F(1,71)=26.15$, $p=2.57E-6$, Interaction $F(1,71)=6.50$, $p=0.013$; Cre- Cohen's $d = 2.040$; Cre+ Cohen's $d = 0.540$; THE: Cre $F(1,71)=.037$, $p=0.847$, Genotype $F(1,71)=1.15E-5$, $p=0.997$, Interaction $F(1,71)=.049$, $p=0.826$. Data points (and numbers) in bars represent biological replicates (animals). Data from panel B are pooled from two separate experiments.

DOI: https://doi.org/10.7554/eLife.46752.002

The following source data and figure supplements are available for figure 1:

**Source data 1.** Source data for *Figure 1A*.
DOI: https://doi.org/10.7554/eLife.46752.005
**Source data 2.** Source data for *Figure 1B*.
DOI: https://doi.org/10.7554/eLife.46752.006
**Figure supplement 1.** TMX-induced restoration of SynGAP protein levels in adult *Syngap1Cre+;+/ls* mice.
DOI: https://doi.org/10.7554/eLife.46752.003
**Figure supplement 1—source data 1.** Source data for *Figure 1—figure supplement 1*.
DOI: https://doi.org/10.7554/eLife.46752.004

data demonstrate that SynGAP re-expression in adulthood improves seizure threshold in *Syngap1* haploinsufficient mice.

Given the behavioral improvements in response to SynGAP re-expression in adult Cre(+)/Lox-Stop Het mice, we explored how this approach impacts neurophysiological correlates of seizure susceptibility. Prior studies have shown through EEG recordings that constitutive *Syngap1* heterozygous knockout mice display frequent high-amplitude interictal spiking (IIS) events in addition to occasional electrographic seizure events (*Ozkan et al., 2014*). Interictal spikes (IIS) are pathological electrical events that reflect seizure susceptibility in patients and may share common mechanisms with ictal events (*Karoly et al., 2016*). We reasoned that because behavioral seizure susceptibility was reversed after SynGAP re-expression in Cre(+)/Lox-Stop Hets, IIS events may also be ameliorated by this therapeutic approach. However, neurophysiological studies have not been performed in *Syngap1* Lox-Stop mice. Therefore, it is unknown if IIS events are present in this line. We performed in vivo neurophysiological recordings in both Cre(+) WT and Cre(+) Lox-Stop Het mice. In addition to left and right sub-cranial EEG electrodes, a depth electrode was lowered into CA1 to acquire a hippocampal local field potential (*h*LFP). A study utilizing chronic recordings in these animals consisted of two phases. Phase one was geared toward identifying putative genotype-dependent differences in neurophysiological signals between Cre(+) WT and Cre(+) Lox-Stop mice. Phase II, in which all

animals in Phase I underwent a procedure to induce Cre recombinase immediately following Phase I studies, was geared toward determining how putative neurophysiological abnormalities in each animal were impacted by restoration of SynGAP levels in Cre(+) Lox-Stop Het mice. In Phase I recordings, we observed frequent high amplitude IIS events that generalized across the three recording channels in Lox-Stop mice (*Figure 2A*; *Figure 2—figure supplement 1*). This finding demonstrates that IIS is a reproducible phenotype across different strains of mice with *Syngap1* haploinsufficiency. We quantified the frequency of IIS events during wakefulness in each genotype and found that spiking events were ~50 fold more frequent in mutants compared to WT controls (*Figure 2B*; *Figure 2— source data 1*). We also observed that high-amplitude IIS events in Lox-Stop animals exhibited state-dependence during Phase one recordings, with an ~18 fold higher frequency of these events observed during sleep compared to wakefulness (*Figure 2A,C*; *Figure 2—source data 2*). We took advantage of the ability to re-express SynGAP protein in Lox-Stop mice to investigate how this strategy impacted IIS events. Remarkably, recordings during Phase II revealed that SynGAP re-expression in adult Lox-Stop mice eliminated IIS activity during wakefulness and sleep (*Figure 2A–B*). These data demonstrate that SynGAP protein dynamically regulates cellular mechanisms that govern systems-level excitability in the mature mouse forebrain.

EEG waveforms are emerging as potential endpoints in clinical research studies for quantifying the efficacy of novel treatments for brain disorders (*Jeste et al., 2015b*; *Modi and Sahin, 2017*). We were therefore interested in determining if EEG-recorded IIS events in *Syngap1* mice have parallels

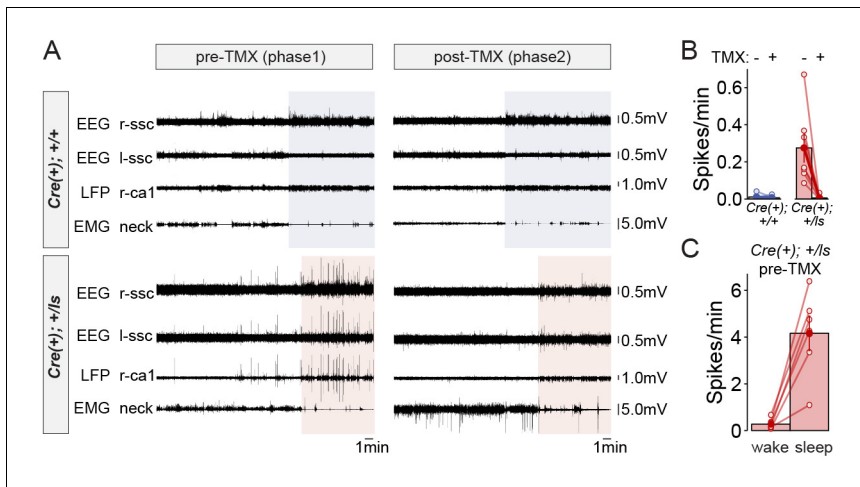

**Figure 2.** Rectification of state-dependent paroxysmal spiking events in *Syngap1* mutants after adult-initiated gene therapy. (A) Representative EEG/LFP traces from a WT [Cre(+); +/+] and *Syngap1* heterozygous mutant mouse [Cre(+); +/ls]. After initial recordings (pre-TMX), all animals were injected with TMX. Post-TMX recordings were acquired 30 days after the last TMX injection. TMX rescued low levels of SynGAP protein in +/ls animals (see *Figure 1—figure supplement 1*). Highlighted areas correspond to periods of sleep (see Materials and methods). Phase I and Phase II recordings are from the same animals. (B) Frequency of spiking events observed in the hippocampal LFP channel during the wake phase (i.e. non-highlighted areas in panel A) from both pre- and post-TMX recording sessions in each animal. Two-way repeated measures ANOVA.:Main genotype effects: $F_{(1,11)}$ =10.1, p=0.00879, Main TMX effects: $F_{(1,11)}$=12.088, p=0.00518. Interaction between genotype and TMX: $F_{(1,11)}$ =9.777, p=0.00963. Cre(+);+/+ n = 6, Cre(+);+/ls n = 7. (C) Comparison of the spiking frequency from the *h*LFP channel in Cre(+);+/ls mice during wake and sleep before TMX injections, paired-*t* test t(5)=-5.6007, p=0.002507 (n = 5). Data points in plots represent biological replicates (animals).
DOI: https://doi.org/10.7554/eLife.46752.007

The following source data and figure supplement are available for figure 2:

**Source data 1.** Source data for *Figure 2B*.
DOI: https://doi.org/10.7554/eLife.46752.009
**Source data 2.** Source data for *Figure 2C*.
DOI: https://doi.org/10.7554/eLife.46752.010
**Figure supplement 1.** Generalization of high-amplitude spikes across the forebrain.
DOI: https://doi.org/10.7554/eLife.46752.008

in humans expressing pathogenic *SYNGAP1* variants. While IIS events are commonly observed in both humans and animal models expressing genetic risk variants linked to epilepsy, worsening of these events during sleep is uncommon and is a symptom of a distinct cluster of epilepsy syndromes, such as electrical status epilepticus in sleep (ESES)/continuous spikes and waves during slow sleep (CSWS) (*Gencpinar et al., 2016*). Therefore, because sleep-dependent worsening of IIS events was observed in *Syngap1* mice, we searched for evidence of sleep-dependent worsening of paroxysmal spiking events in MRD5 patients. To do this, we mined data from a *SYNGAP1* patient registry, which has been used to uncover previously unreported phenotypes in this NDD population (*Michaelson et al., 2018*). In this database, we found 11 entries that included a relatively complete medical history, including an EEG report from a neurologist and a detailed genetic report defining the *SYNGAP1* variant. Each of the selected patients was confirmed to have a severe *SYNGAP1* variant expected to cause genetic haploinsufficiency (i.e. heterozygous knockout caused by either a nonsense variant or a frameshift caused by an InDel). In each of these unique patient entries, there were clear references to IIS activity (*Supplementary file 3*). However, there were six unique entries where the neurologist noted that spike-related events worsened during sleep. In one patient (#5569), the neurologist noted that epileptiform activity was nearly constant during light sleep. In another (#1029), IIS was observed in 30% of recorded sleep. Moreover, we obtained EEG recordings from two additional patients, S3-060 and S3-080, confirmed to express variants consistent with *SYNGAP1* haploinsufficiency. In these cases, we also observed evidence of sleep-dependent worsening of pathological EEG signals (*Figure 3A–D*). These clinical observations indicate that some patients with *SYNGAP1* haploinsufficiency have worsening EEG profiles during sleep, which strengthens the validity of sleep-dependent worsening of IIS in *Syngap1* mouse models.

We next explored other measures of cognitive function impacted by SynGAP re-expression in adulthood. To date, only spontaneous alternation has been used to determine the impact of adult reversal of *Syngap1* pathogenicity on changes to cognitive function (*Clement et al., 2012*). This measure of working memory was not improved after protein restoration in adult *Syngap1* Het mice. To expand cognitive-based testing of adult gene-restoration, we employed contextual fear conditioning, which requires animals to associate a novel context (e.g. the training environment) with an adverse event (e.g. mild foot-shock). Memory is inferred from a behavior change, such as freezing and/or immobility, when the animal is placed back in the training environment hours or days later.

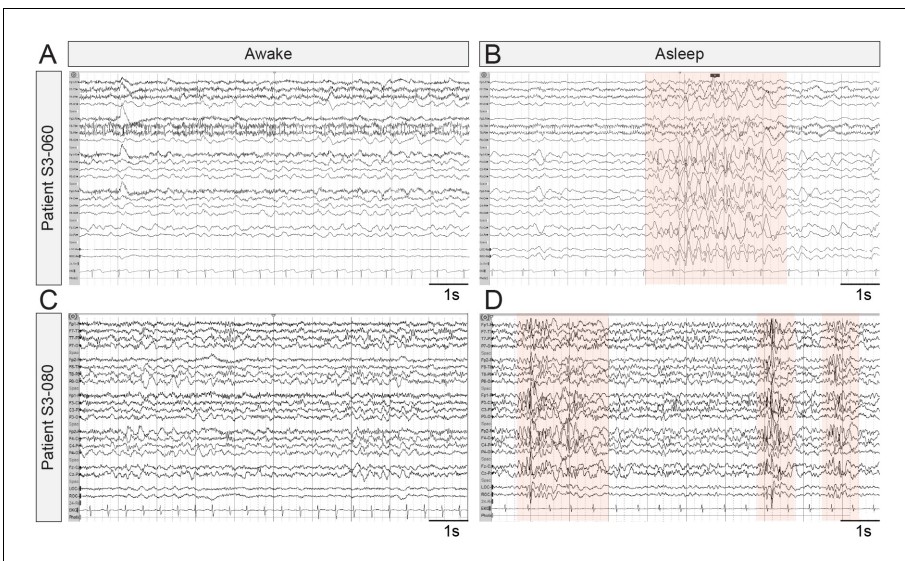

**Figure 3.** Representative EEG recordings taken from *SYNGAP1* patients during wake and sleep. Ten second epochs of electroencephalograms from patients with SYNGAP1 pathogenic variants. (**A**) Patient S3-060 while awake (**B**) Patient S3-060 while asleep (**C**) Patient S3-080 while awake (**D**) Patient S3-080 while asleep. Shaded areas indicate bursts of generalized epileptiform activity.
DOI: https://doi.org/10.7554/eLife.46752.011

We, and others, have previously reported that *Syngap1* Het mice have normal expression of contextual fear 1 day (1d) after training (**Guo et al., 2009**; **Muhia et al., 2010**). However, *Syngap1* Het mice display disrupted remote contextual memory because they have increased immobility when tested one month after training (**Ozkan et al., 2014**). Thus, we focused on remote memory in this study. For these experiments, we used activity suppression ratios (ASR) to quantify behavioral performance rather than freezing. We did this because ASR is a well-validated measure of context fear and is reported to be a more reliable measure of context memory in mice with suspected differences in basal levels of detectable freezing (**Anagnostaras et al., 2000**). Indeed, we have observed that the inclusion of the Cre-ER allele can cause intermittent, subtle twitching in mice during obvious bouts of freezing. This twitching behavior confounds automated measures of freezing. Here, we report that two different *Syngap1* Het strains (i.e. constitutive *Syngap1* Het mice and *Syngap1* Lox-Stop mice) exhibit a deficit in contextual memory deficit when tested 26-30d after training (**Figure 4A**; **Figure 4—source data 1**; **Figure 4B**; **Figure 4—source data 2**). The first few days after contextual fear conditioning is marked by remote memory consolidation (**Frankland et al., 2004**), a systems-level process that involves communication between the hippocampus and the cortex and is believed to convert memory traces into a more permanent form of storage or maintenance. A selective deficit at remote time points is indicative of a systems memory consolidation deficit in *Syngap1* mice.

Upon further testing, we discovered that retrieval 1d after training prevented 26-30d remote memory deficits in *Syngap1* Hets (**Figure 4C**; **Figure 4—source data 3**). The retrieval-induced protection of remote memory in adult *Syngap1* animals suggested that the biological processes that promote systems consolidation may not be permanently damaged by heterozygosity of this gene during development. Therefore, we hypothesized that adult re-expression of SynGAP protein would improve the function of brain circuits that support remote memory. To do this, we again used the *Syngap1* Lox-Stop Het animal model that enables whole-body SynGAP protein re-expression beginning in adulthood. We trained adult offspring resulting from this cross in contextual fear conditioning and tested memory ~1 month after treatments with or without TMX (**Figure 4D**). In TMX(-) mice, we observed a strong main effect on genotype (i.e. the presence of Lox-Stop cassette; p=9.28E-10), but not with the presence of Cre-ER (p=0.864), which further supports the idea that *Syngap1* heterozygosity impacts remote memory (**Figure 4E**; **Figure 4—source data 4**). We observed a relatively weak statistical interaction between Cre-ER and Lox-Stop (p=0.011). Importantly, pairwise comparisons among all four groups indicated that both Cre(-) and Cre(+) *Syngap1* Hets performed significantly differently compared to corresponding *Syngap1* WT controls (**Figure 4E**; **Supplementary file 4**; **Figure 4—source data 5**). There was also no difference in performance between Cre(-) and Cre(+) *Syngap1* Hets (**Figure 4E**; **Supplementary file 4**), indicating that Cre remained sufficiently inactive in the absence of TMX in these mice.

In TMX(+) studies (**Figure 4F**), we again observed a strong main effect on genotype (p=1.48E-6), but not with the presence of Cre-ER (p=0.891). In contrast to TMX(-) studies, here we observed a highly significant interaction between Cre-ER and Lox-Stop (p=2.59E-4), which suggested that SynGAP re-expression altered activity suppression levels. Pairwise comparisons of all four genotypes indicated that elevated ASR levels normally found in *Syngap1* haploinsufficient mice were reversed back to WT levels by TMX injections. For example, TMX-treated Cre(+) *Syngap1* Het mice were no different from corresponding Cre(+) WT *Syngap1* mice (**Figure 4F**; **Supplementary file 4**). Moreover, the ASR in these mice was significantly reduced compared to Cre(-) *Syngap1* Lox-Stop Hets [i.e. mice with preserved *Syngap1* pathogenicity), consistent with adult reversal of the remote memory deficit (**Figure 4F**; **Supplementary file 4**). It is important to note that while we did not observe any main effects of Cre-ER in either TMX(-) or TMX(+) studies, there is a trend for Cre-ER to increase ASR in WT mice (i.e. mice without the Lox-Stop allele) across both experiments. However, we conclude that this trend did not confound the interpretation of how gene restoration impacted behavioral performance in this experiment. To the contrary, the trend for Cre-ER to increase ASR in WT mice was actually reversed when comparing Cre(-) and Cre(+) Lox-Stop mice (**Figure 4F**). We interpreted this as further evidence that SynGAP re-expression improved performance in the memory task. Taken together, these data support the interpretation that SynGAP re-expression in adulthood rescues remote contextual memory deficits in *Syngap1* haploinsufficient mice.

Next, we directly tested the idea that *Syngap1* gene restoration improves the function of neural circuits that promote systems-memory consolidation. Hippocampal oscillations are linked to a range of cognitive functions, including various mnemonic processes associated with systems memory

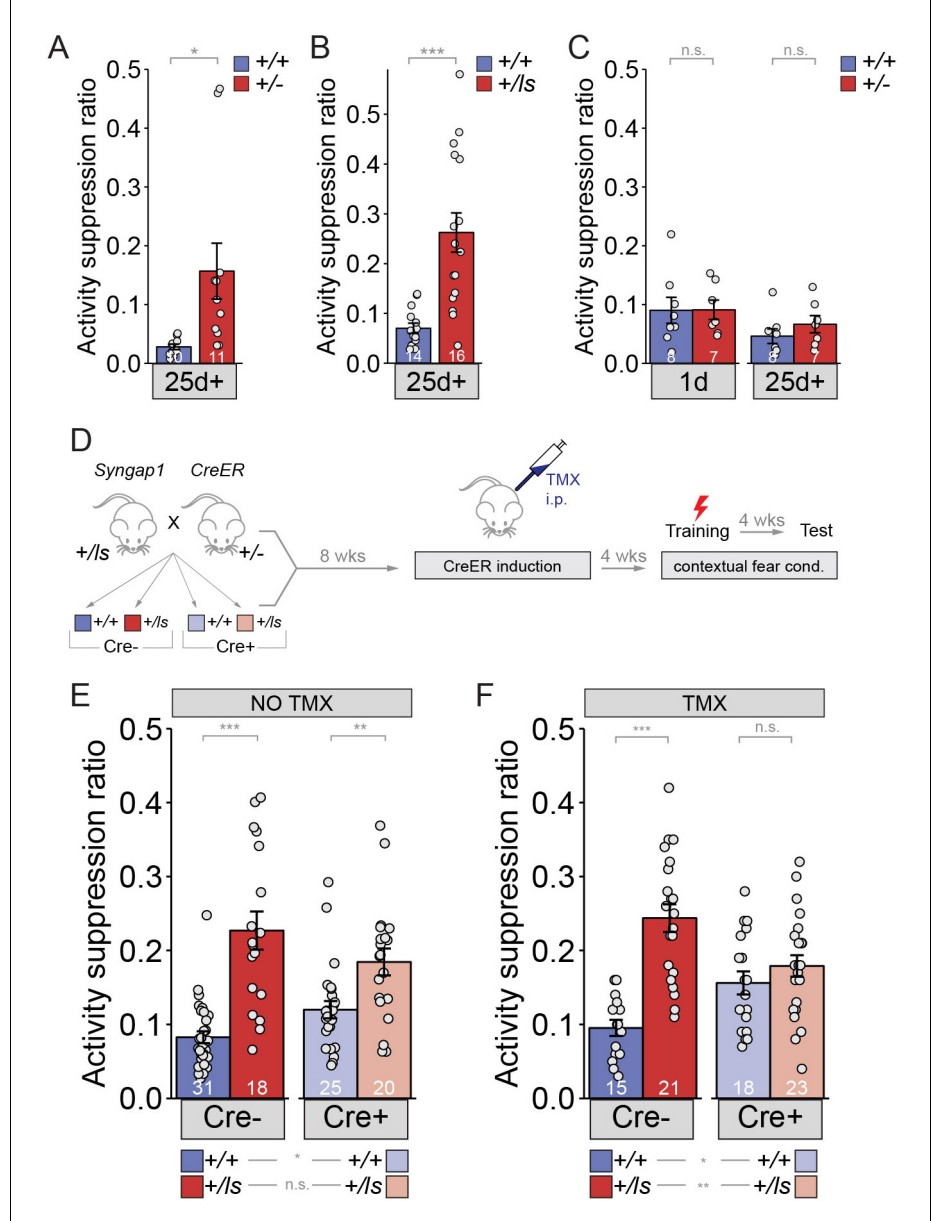

**Figure 4.** Long-term memory can be improved in adult mice with *Syngap1* pathogenicity. (**A**) *Syngap1*$^{+/+}$ and *Syngap1*$^{+/-}$ ± were trained in the remote contextual fear conditioning paradigm and tested one month later for activity suppression levels. Activity of the *Syngap1*$^{+/-}$ was suppressed significantly less than that of the *Syngap1*$^{+/+}$ group indicating compromised remote memory for the mutant group. Unpaired *t* test (t(19)=-2.567, p=0.019). Cohen's d = 1.150. (**B**) *Syngap1*$^{+/+}$ and *Syngap1*$^{+/ls}$ mice were trained in the contextual fear conditioning paradigm and tested one month later for activity suppression levels. Activity of the *Syngap1*$^{+/ls}$ group was suppressed significantly less than that of the *Syngap1*$^{+/+}$ group indicating compromised remote memory for the mutant group. Wilcoxon rank sum test W = 19, p=2.82E-5, Cohen's d = 1.676. (**C**) *Syngap1*$^{+/+}$ and *Syngap1*$^{+/-}$ were tested, firstly, 1d after training, followed by another testing one month later. Activity suppression levels were not significantly different between the groups for either testing (unpaired t test,1-day t(13)=-0.033, p=0.974; 26 days t(13)=-1.068, p=0.305). (**D**) Experimental schematic depicting the breeding strategy for generation of Cre-inducible *Syngap1*$^{Cre+;+/ls}$ mice and Cre induction with TMX treatment for restoration of *Syngap1* expression and subsequent remote fear conditioning testing. (**E–F**) *Syngap1*$^{Cre-;+/+}$, *Syngap1*$^{Cre-;+/ls}$, *Syngap1*$^{Cre+;+/+}$, and *Syngap1*$^{Cre+;+/ls}$ mice were run in the remote contextual fear conditioning paradigm without (**E**) and with (**F**) TMX administration. Activity suppression values from mice without TMX administration (**No TMX**) were assessed (2-factor ANOVA: Main Effects-Cre F(1,90)=0.030, p=0.864, Genotype F(1,91)=46.78, p=9.28E-10, Interaction F(1,91)=6.81, p=0.011; Cre- Cohen's d = 1.725, Cre+ Cohen's d = 0.910. **With TMX administration** (2-factor ANOVA:
*Figure 4 continued on next page*

*Figure 4 continued*

Main Effects- Cre F=(1,73)=0.019, p=0.891, Genotype F(1,73)=27.49, p=1.48E-6, Interaction F(1,73)=14.75, p=2.59E-4; Cre- Cohen's d = 2.167). Data points (and numbers) in bars represent biological replicates (animals). Data from panels E-F are pooled from at least two separate experiments.

DOI: https://doi.org/10.7554/eLife.46752.012

The following source data is available for figure 4:

**Source data 1.** Source data for *Figure 4A*.
DOI: https://doi.org/10.7554/eLife.46752.013
**Source data 2.** Source data for *Figure 4B*.
DOI: https://doi.org/10.7554/eLife.46752.014
**Source data 3.** Source data for *Figure 4C*.
DOI: https://doi.org/10.7554/eLife.46752.015
**Source data 4.** Source data for *Figure 4E*.
DOI: https://doi.org/10.7554/eLife.46752.016
**Source data 5.** Source data for *Figure 4F*.
DOI: https://doi.org/10.7554/eLife.46752.017

consolidation (*Colgin, 2016*). Therefore, we recorded hippocampal oscillations through an LFP electrode implanted in CA1. We recorded these signals in mice both before and after *Syngap1* gene restoration, which enabled us to assess how this therapeutic strategy impacted hippocampal function in each animal. Theta rhythms were clearly observable in WT mice during both recording phases (*Figure 5A*). However, in recordings from Lox-Stop mice during Phase I, theta rhythms often

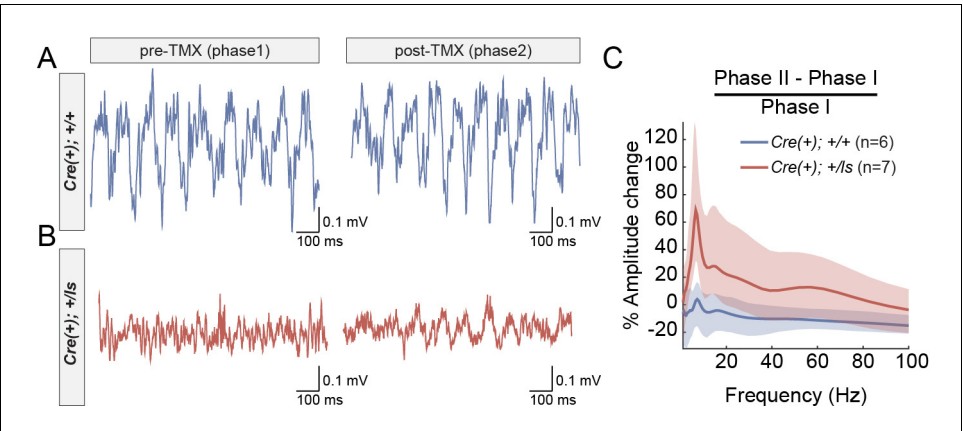

**Figure 5.** Increased amplitude of theta oscillations after SynGAP re-expression in adult *Syngap1* mutant mice. (A–B) CA1 LFP traces from a WT (**A**) and a *Syngap1* mutant (**B**) mouse during Phase I and Phase II sessions. (**C**) Grand average of within-subjects changes in signal amplitude across the full spectrum of hippocampal rhythms. The amplitude change was normalized by the average amplitude during Phase I sessions. The shaded areas represent 95% bootstrapped confidence intervals. Significant increases in amplitude in Phase II were detected in the 6–12 Hz theta range (Permutation test: p=0.0128, 5000 shuffles). N's are biological replicates (animals). Legends for Figure Supplements.

DOI: https://doi.org/10.7554/eLife.46752.018

The following source data and figure supplements are available for figure 5:

**Figure supplement 1.** Amplitude of theta oscillations in each mouse during Phase I and Phase II recording sessions.
DOI: https://doi.org/10.7554/eLife.46752.019
**Figure supplement 2.** Effect of genotype, but not phase, on horizontal activity during neurophysiological recordings.
DOI: https://doi.org/10.7554/eLife.46752.020
**Figure supplement 2—source data 1.** Source data for *Figure 5—figure supplement 2*.
DOI: https://doi.org/10.7554/eLife.46752.021

appeared less robust during periods of free exploration compared to WT littermates (*Figure 5B*; *Figure 5—figure supplement 1*). After SynGAP re-expression, theta oscillations in these animals increased compared to pre-injection baseline recordings (*Figure 5B*). Across the entire Lox-Stop population, we found that theta-range signal amplitude was significantly enhanced by raising Syn-GAP protein levels in this population of adult Lox-Stop mice (*Figure 5C*), which is a finding consistent with improved performance in memory tasks considering that increased theta rhythm expression has been shown to correlate with better memory performance.

Theta rhythms in mice are modulated by running speed (*Gereke et al., 2018*) and it has been reported that *Syngap1* Het mice are hyperactive in some, but not all, contexts (*Kilinc et al., 2018*). Therefore, we tracked locomotor activity in all recorded mice across both sessions to determine if phase-specific changes in activity levels may have contributed to changes in theta amplitude (*Figure 5—figure supplement 2*; *Figure 5—figure supplement 2—source data 1*). Similar to past studies, Cre(+) Lox-Stop Het mice were hyperactive compared Cre(+)WT controls as evidenced by a significant main effect of genotype (i.e. group) on distance traveled (p=0.002). However, we did not observe an effect of phase (p=0.692), or an interaction between phase and genotype in this analysis (p=0.085). Therefore, the phase-independent increases in locomotor activity in Het mice are unlikely to explain the phase-specific changes to theta rhythms in these animals.

## Discussion

The principal finding in this study is that genetic reversal of *Syngap1* pathogenicity in adult mice reverses deficits in remote memory, lowers seizure threshold, re-balances neural excitability, and improves hippocampal/cortical function. On the surface, these exciting results are somewhat paradoxical given that we have previously found that *Syngap1* pathogenicity causes hardwired impairments in the development of forebrain circuits that mediate a subset of previously tested cognitive functions (*Clement et al., 2012*; *Aceti et al., 2015*; *Ozkan et al., 2015*). These previous studies indicated that impaired assembly of forebrain circuits contribute to life-long impairments in brain function because a set of behaviors were shown to be resistant to adult re-activation of SynGAP protein expression. How then can these current findings be reconciled with past findings demonstrating that certain behavioral paradigms are not sensitive to adult re-expression of SynGAP protein? The most parsimonious explanation is that not all behavioral impairments common to the *Syngap1* endophenotype are governed by hardwired circuit changes, which in turn reflect altered neurodevelopmental processes. Some behavioral impairments may be caused by altered 'real-time' neuronal signaling that is not restricted to defined periods of neurodevelopment (*Zoghbi and Bear, 2012*). Elegant work in *Shank3* mouse models directly supports this interpretation (*Mei et al., 2016*). Feng and colleagues demonstrated that adult reversal of *Shank3* pathogenicity improved some non-cognitive and non-seizure-related behavioral phenotypes, while other independent phenotypes remained completely impaired even though protein levels were restored to WT levels. Restoring Shank3 protein levels early in development prevented the emergence of adult reversal-resistant behavioral impairments, indicating that low expression of germline Shank3 protein disrupts both developmental and non-developmental cellular processes.

Evidence in the literature indicates that *Syngap1* may have unique functions during both development and adulthood, which supports an evolving interpretation of SynGAP function in the brain. For instance, *Syngap1* heterozygosity causes a transient increase in baseline postsynaptic function in CA1 Schaffer collateral synapses during postnatal development (*Clement et al., 2012*). This is a true developmental impact of *Syngap1* pathogenicity because inducing pathogenicity in adulthood does not induce a change in baseline excitatory synapse function in hippocampal synapses. Interestingly, baseline excitatory function of this pathway normalizes by adolescence, as evidenced by several groups (*Kim et al., 2003*; *Komiyama et al., 2002*), including our own (*Clement et al., 2012*), reporting normal postsynaptic baseline function in this synapse in mature animals. However, there is a dramatic impairment in LTP stability at these synapses in adult *Syngap1* mice (*Kim et al., 2003*; *Komiyama et al., 2002*). This function of *Syngap1* appears unrelated to neurodevelopment because LTP can be completely rescued after adult re-expression of low SynGAP protein levels (*Ozkan et al., 2014*). Thus, *Syngap1* exhibits both developmental and non-developmental functions within the same synapses in the hippocampus. As a consequence, not all deficits related to neuronal function

in the adult *Syngap1* Het mouse brain can be attributed to hardwired circuit damage caused by impaired neurodevelopment.

The real-world impact of our current findings is two-fold. First, therapies that improve the expression and/or function of SynGAP protein may be beneficial to MRD5 patients of all ages. Adult re-expression of SynGAP appears to correct neurophysiological imbalances within circuits that predispose the brain to seizure generation and certain forms of cognitive impairment. Therefore, the impact of therapies related to elevated SynGAP levels in adult patients could be significant. It remains unclear if adult re-expression of SynGAP corrects seizure susceptibility and memory impairments through a common mechanism. Future studies will be necessary to determine the molecular and cellular mechanisms that are recovered in the forebrain after SynGAP re-expression. Moreover, it will be critical to link these recovered molecular and cellular processes to circuit-level processes that directly contribute to behavioral memory and seizure. This type of experimental strategy may help to elucidate novel neural circuit correlates of behavioral alterations and excitability impairments associated with NDDs. However, early therapeutic intervention should remain the primary NDD treatment strategy. Treatments during development would protect the brain from both hardwired circuit damage caused by disruptions to critical neurodevelopmental processes (*Clement et al., 2012*; *Aceti et al., 2015*; *Clement et al., 2013*) and 'real-time' age-independent neurophysiological imbalances described in this study.

The second real-world implication of this study is that it may have identified a candidate translatable biomarker that can serve as an endpoint for efficacy testing of novel treatments for MRD5 patients. Our data indicate that IIS events appear to worsen during sleep in both humans and mice with *SYNGAP1/Syngap1* haploinsufficiency. Sleep-linked worsening of IIS is a hallmark of epilepsy syndromes with significant cognitive impairment (*Gencpinar et al., 2016*). Similar to what we report here, a recent, comprehensive analysis of a cohort of *SYNGAP1* patients observed a sleep-dependent increase in pathological EEG waveforms in approximately half of the children (*Vlaskamp et al., 2019*). We found that SynGAP re-expression in adult mutant mice eliminated these pathological events, which are also detected by EEG electrodes, during both wakefulness and sleep. Considering that this treatment strategy also improved measures of behavioral seizure and memory, the severity of IIS during sleep may be a predictor of generalized cortical impairment associated with *SYNGAP1* pathogenicity. If a relationship between the severity of state-dependent IIS and cognitive impairment is observed in patients, then these signals may be useful as biomarkers for improved cortical function in translational studies aimed at identifying effective therapeutic strategies for *SYNGAP1* patients.

## Materials and methods

### Animals and behavior

*Syngap1* constitutive (*Syngap1*$^{+/-}$; RRID: MGI:3576223) and reversal (*Syngap1*$^{+/ls}$; RRID:MGI:5469866) mice were constructed and maintained as previously described (*Ozkan et al., 2014*). Inducible CAGG-Cre-ER male mice (RRID: MGI:2182767) were purchased from Jackson Laboratories (JAX stock #004682) and crossed to female *Syngap1*$^{+/ls}$ for adult reversal studies. Male and female experimental mice were utilized for all studies. Animals were group housed (n = 4/cage) by sex and otherwise randomly assigned to cages yielding mixed genotypes. Moreover, each cohort of mice was represented by multiple litters. Animals were kept on a normal light-dark cycle and had free access to food and water. Animal experiments were conducted according to protocols submitted to, and approved by, Scripps Research (Protocol #15–037 and #15–038) and the Baylor College of Medicine (Protocol #AN5585) Institutional Animal Care and Use Committees.

Mice (PND90-120) were handled for several minutes on three separate days prior to commencement of behavioral testing. Tails were marked for easy identification and access from home cages during testing. Various cohorts of mice were utilized in this study consisting of one or two cohorts of naïve *Syngap1*$^{+/ls}$ mice for the TMX(-) flurothyl-induced seizure and remote fear conditioning experiments, respectively and two other cohorts of *Syngap1*$^{+/ls}$ mice for both the fear conditioning and seizure TMX(+) experiments. Several mice from the TMX(+) inducible Cre cohorts were unhealthy or died prior to testing, so we ran two cohorts for the TMX(+) experiments. One naïve cohort of non-inducible *Syngap1* constitutive or lox/stop mice were used for the other fear conditioning experiments in *Figure 4A–C*. Samples sizes for *Syngap1*-related studies in our lab have been estimated

using GPower 3.0. Estimates for effect size and variance of behavioral data were based on published studies using the *Syngap1* mouse lines (*Clement et al., 2012*; *Aceti et al., 2015*; *Ozkan et al., 2014*; *Guo et al., 2009*). Experimenters were blind to genotype while conducting behavioral tests and during data analysis.

## Cre Induction

TMX (Sigma T5648, St. Louis, MO) was prepared in corn oil containing 10% EtOH to a final TMX dosage of 100 mg/kg, injectable concentration of 20 mg/ml, and volume of 5 ml/kg and administered (intraperitoneal) once a day for five consecutive days starting at PND60.

*Contextual fear conditioning:* A dedicated fear conditioning room in the TSRI Florida Mouse Behavior Core contains four fear conditioning apparati that can be used in parallel. Each apparatus was an acrylic chamber measuring approximately 30 × 30 cm (modified Phenotyper chambers, Noldus, Leesburg, VA). The top of the chamber is covered with a unit that includes a camera and infrared lighting arrays (Noldus, Ethovision XT 11.5, Leesburg, VA) for monitoring of the mice. The bottom of the chamber is a grid floor that receives an electric shock from a shock scrambler that is calibrated to 0.40 mA prior to experiments. The front of the chamber has a sliding door that allows for easy access to the mouse. The chamber is enclosed in a sound-attenuating cubicle (Med Associates) equipped with a small fan for ventilation. Black circular, rectangular and white/black diagonal patterned cues were placed outside each chamber on the inside walls of the cubicles for contextual enhancement. A strip light attached to the ceilings of the cubicles provided illumination. A white noise generator (~65 dB) was turned on and faced toward the corner of the room between the cubicles. The fear conditioning paradigm consisted of two phases, training, followed by testing 1 and 26, or 30 d thereafter. The 4.5 min training phase consisted of 2.5 min of uninterrupted exploration. Two shocks (0.40 mA, 2 s) were delivered, one at 2 min 28 s, the other at 3 min and 28 s from the beginning of the trial. During testing, mice were placed into their designated chambers and allowed to roam freely for 5 min. Immobility durations (s) and activity (distances moved (cm)) during training and testing were obtained automatically from videos generated by Ethovision software. Activity suppression ratio levels were calculated: 0–2 min activity during testing/0–2 min activity during training +testing.

## Flurothyl-induced seizures

Flurothyl-induced seizure studies were performed based on prior studies with some modifications (*Clement et al., 2012*; *Ozkan et al., 2014*; *Dravid et al., 2007*). Briefly, experiments were conducted in a chemical fume hood. Mice were brought to the experimental area at least 1 hr before testing. To elicit seizures, individual mice were placed in a closed 2.4 L Plexiglas chamber and exposed to 99% Bis (2,2,2-triflurothyl) ether (Catalog# 287571, Sigma-Aldrich, St. Louis, MO). The flurothyl compound was infused onto a filter paper pad, suspended at the top of the Plexiglas chamber through a 16G hypodermic needle and tube connected to a 1 ml BD glass syringe fixed to an infusion pump (KD Scientific, Holliston, MA, USA, Model: 780101) at a rate of 0.25 ml/min. The infusion was terminated after the onset of a hind limb extension that usually resulted in death. Cervical dislocation was performed subsequently to ensure death of the animal. Seizure thresholds were measured as latency (s) from the beginning of the flurothyl infusion to the beginning of the first myoclonic jerk (1 st clonus), then to generalized tonic/clonic seizure (T/C), and finally to total hind limb extension (THE).

## Immunoblotting

Western blot analysis was performed on protein lysates extracted from the hippocampi of adult mice and dissected in ice-cold PBS containing Phosphatase Inhibitor Cocktails 2 and 3 (Sigma-Aldrich, St. Louis, MO) and Mini-Complete Protease Inhibitor Cocktail (Roche Diagnostics) and immediately homogenized in RIPA buffer (Cell Signaling Technology, Danvers, MA). Sample protein concentrations were measured (Pierce BCA Protein Assay Kit, Thermo Scientific, Rockford, IL), and volumes were adjusted to normalize microgram per microliter protein content. 10 μg of protein per sample were loaded and separated by SDS-PAGE on 4–15% gradient stain-free tris-glycine gels (Mini Protean TGX, BioRad, Hercules, CA), transferred to low fluorescence PVDF membranes (45 μm) with the Trans-Blot Turbo System (BioRad). Membranes were blocked with 5% powdered milk in

buffer and probed with pan-SynGAP (1:10,000, PA1-046, Pierce/Thermo Scientific) overnight at 4°C and HRP-conjugated anti-rabbit antibody (1:2,500, W4011, Promega) for 1 hr at room temperature followed by ECL signal amplification and chemiluminescence detection (SuperSignal West Pico Chemiluminescent Substrate; Thermo Scientific, Rockford, IL). Blot band densities were obtained using the Alpha View imaging system (Alpha Innotech). SynGAP levels of immunoreactivity were assessed by densitometric analysis of generated images with ImageJ. Values were normalized to total protein levels obtained from blots prior to antibody incubations.

## Video-EEG recordings from mice

Experiments were carried out on Cre(+)/*Syngap1* Lx-ST heterozygous mice and littermate Cre(+)/ *Syngap1* WT controls. Experimenters were blind to the genotypes. Animals were bred at The Scripps Research Institute and experimental mice transferred to Baylor College of Medicine at ~4–5 weeks of age. Animals at 11 weeks of age were secured on a stereotaxic frame (David Kopf) under 1–2% isoflurane anesthesia. Each mouse was prepared under aseptic condition for the following recordings: Teflon-coated silver wires (bare diameter 127 µm, A-M systems) were implanted bilaterally in the subdural space of the somatosensory cortex (*Paxinos and Franklin, 2001*) (0.8 mm posterior, 1.8 mm lateral to bregma) with reference to the midline over the cerebellum) for cortical EEG as well as in neck muscles for electromyogram recordings to monitor mouse activity. An additional electrode constructed with Teflon-coated tungsten wire (bare diameter 50 µm, A-M systems) was stereotaxically implanted in the CA1 of the hippocampus (*Paxinos and Franklin, 2001*) (1.9 mm posterior, 1.0 mm lateral, and 1.3 mm ventral to bregma) with reference to the ipsilateral corpus callosum for local field potential recordings. All electrode wires and the attached miniature connector sockets were fixed to the skull by dental cement. After 2 weeks of post-surgical recovery, mice received Phase I video-EEG recordings (2 hr per day for 3 days). Signals were amplified (100x) and filtered (bandpass, 0.1 Hz - 1 kHz) by 1700 Differential AC Amplifier (A-M Systems), then digitized at 2 kHz and stored on disk for off-line analysis (DigiData 1440A and pClamp10, Molecular Devices). Time-locked mouse behavior was recorded by ANY-maze tracking system (Stoelting Co.). In addition, manual ON/OFF camcorder was used to monitor the behavior at higher resolution. Beginning the day after Phase I recordings of video-EEG, all mice received daily injections of Tamoxifen (*as above*) for 5 days. One month later, these mice were subjected to Phase II video-EEG recordings (three 2 hr sessions over 3 days) under the same settings as in Phase I. At the end of the experiments, mice were euthanized, and hippocampi were dissected to determine the efficacy of SynGAP protein re-expression.

## High-Amplitude interictal spike quantification

Axograph3 or pClamp10 software (Molecular Devices, San Jose, CA) was used to detect high-amplitude spiking events. The threshold was set at +1 mV in the CA1 depth channel for all animals and all events that exceeded the threshold were logged. Events were occasionally rejected as 'non-physiological'. These rejected signals were identified by their usually high signal amplitude and time-locked peaks in more than one channel and their atypical shape compared to paroxysmal spikes. Behavioral epochs were segregated into sleep and wake phases. Sleep was inferred from an abrupt quieting of the EMG signal and confirmation of immobility from time-locked videos. Experimenters were blind to genotype while conducting spike quantification analyses.

## Analysis of hippocampal oscillations

Analyzes focused on the first 10 min of each recording session for each mouse to account for experience-dependent changes in mouse hippocampal rhythms (*Gereke et al., 2018*). Videos of mouse behavior and EMG recordings were used to verify that mice were engaged in active locomotion during this period, considering that theta oscillations occur most prominently in the hippocampus during voluntary movement (*Vanderwolf, 1969*). Time-resolved amplitudes of CA1 LFP recordings were estimated using a complex Morlet wavelet transform with a width parameter of six periods, evaluated at 50 frequencies logarithmically spaced between 1 and 100 Hz (*Tallon-Baudry et al., 1998*). Wavelets were normalized such that their area under the curve was equal to one (*Liu et al., 2007*). The magnitude of theta (6–12 Hz) oscillations across time was quantified using the ratio of theta to delta (1–4 Hz) power. The power ratio was smoothed by a moving average time window of

1 s. The one-second time window with the highest theta-delta power ratio for each mouse in each Phase was selected for the representative CA1 LFP recordings shown in *Figure 5A & B* and *Figure 5—figure supplement 1*. To account for the variability of theta amplitudes across mice (see *Figure 5—figure supplement 1*), amplitude changes between Phase I and Phase II sessions were calculated within each mouse and then normalized by the corresponding amplitude during Phase I sessions (*Figure 5C*). Experimenters were blind to genotype while conducting LFP analyses.

## Human subject data

Collection of human subject data was reviewed and approved by Hummingbird (Study # 2016–57-SYNGAP) and Baylor College of Medicine (Study #H-30480 and #H-41411) Institutional Review Boards.

## Collection and analysis of data from the retrospective SYNGAP1 natural history study registry

The *SYNGAP1* Patient Registry (*Michaelson et al., 2018*; https://syngap1registry.iamrare.org) is funded through the National Organization of Rare Disorders. This study was Institutional Review Board (IRB)-approved and meets all relevant ethical regulations for protections for human subjects. It is actively managed by a board of trustees comprised of a team of seven stakeholders, including parents with affected children, clinician-scientists that care for MRD5 patients, and neurobiologists that study the gene. The *SYNGAP1* (MRD5) Natural History Study Registry is a retrospective longitudinal web-based observational natural history study. Parents or guardians provided informed consent prior to depositing medical history data into the registry. Participants with *SYNGAP1* (MRD5) will be followed throughout the course of their lives with either the participant or authorized respondents contributing data at varying intervals throughout the course of the study. Initially, when a new patient is registered, data are collected on demographics, quality of life, medical history including genetic reports, disease phenotypes, event episodic data, retrospective data, participant-review of systems, medication, and diagnostic data. Each registrant is given a unique identifier to facilitate anonymization of patient data. Initial data collection is done through a series of questionnaires, including a survey of sensory and sensory-related issues. The structure of the database and all questionnaires were reviewed and approved by the members of the Board of Trustees.

To acquire information about EEG during sleep and wakefulness in the *SYNGAP1* patient population, the registry database was queried for all entries that: 1) had a detailed genetic report that confirmed the presence of a severe *SYNGAP1* loss-of-function variant likely to induce genetic haploinsufficiency; 2) had at least a narrative report from a neurologist that discussed the findings of an EEG study. *Supplementary file 3* summarizes each patient-specific *SYNGAP1* variant and findings from their EEG study. A subset of EEG reports contained information about the patients' overall clinical presentations and medications at the time of their EEG study. This information was included in *Supplementary file 3* where appropriate.

## Human EEG

The parents of patients S3-060 and S3-080, which are distinct patients from those represented in *Supplementary file 3*, provided written informed consent according to a protocol approved by the Baylor College of Medicine Institutional Review Board. The medical record and genetic reports were reviewed by a board-certified neurologist. Patient S3-060 was determined to harbor a pathogenic *SYNGAP1* variant [c.1154-1161del (p.S385fs)] as was patient S3-080 [c.3190 C > T (p.Q1064X)]. Each patient had a scalp electroencephalogram with a minimum of 21 electrode recordings in a standard 10–20 distribution and a minimum of 45 min of recording which were reviewed by board-certified neurophysiologists as part of clinical care. The electroencephalograms were further manually reviewed by a board-certified neurologist (JLH) and representative epochs were captured.

## Statistics

Statistical tests for mouse seizure and fear conditioning data were conducted using SPSS and R packages importing Excel files containing raw data. Based on our prior experience with these behaviors, a general parametric statistical approach was used to assess data sets. Unpaired t tests for experiments with two groups comparing genotypes and two-factor ANOVAs with Bonferroni pair-

wise comparisons for experiments with four groups assessing contingencies of Cre and genotype factors were performed to assess activity suppression ratio values associated with fear conditioning. MANOVAs were performed to assess Cre/genotype contingencies for the multiple levels of factors associated with seizure threshold (1$^{st}$ clonic, T/C, and THE). In behavioral studies, we included *Cohen's d* for *Syngap1*$^{+/+}$ vs. *Syngap1*$^{+/(-)or(ls)}$ as a measure of effect size when means were found to be significant between relevant groups. Explorations of data sets showed reasonably normal distributions and homogeneity of variances as assessed by Levene's test for equality of variances as well as Box's test for equality of covariance matrices. A nonparametric permutation test (5000 shuffles) was used to assess differences in oscillatory activity.

## Acknowledgements

This work was supported in part by NIH grants from the National Institute of Mental Health [MH096847 and MH108408 (GR), MH105400 (CAM), MH102450 (LLC)], the National Institute of Neurological Disorders and Stroke [NS064079 (GR), NS100738 (JT)], and the National Institute on Drug Abuse [DA034116 and DA036376 (CAM), T32DA01892 (EH)]. JLH is supported by a National Institute of Neurological Disorders and Stroke Mentored Clinical Scientist Research Career Development Award [NS091381] and the Robbins Foundation. The video-EEG experiments were performed by the IDDRC Neuroconnectivity Core at Baylor College of Medicine.

## Additional information

### Competing interests

Laura L Colgin: Reviewing editor, *eLife*. The other authors declare that no competing interests exist.

### Funding

| Funder | Grant reference number | Author |
| --- | --- | --- |
| National Institute on Drug Abuse | T32DA01892 | Ernie Hwaun |
| National Institute of Neurological Disorders and Stroke | Mentored Clinical Scientist Research Career Development Award NS091381 | Jimmy Lloyd Holder Jr |
| Robbins Foundation | | Jimmy Lloyd Holder Jr |
| National Institute of Neurological Disorders and Stroke | NS100738 | Jianrong Tang |
| National Institute of Mental Health | MH102450 | Laura L Colgin |
| National Institute of Mental Health | MH105400 | Courtney A Miller |
| National Institute on Drug Abuse | DA034116 | Courtney A Miller |
| National Institute on Drug Abuse | DA036376 | Courtney A Miller |
| National Institute of Mental Health | MH108408 | Gavin Rumbaugh |
| National Institute of Mental Health | MH096847 | Gavin Rumbaugh |
| National Institute of Neurological Disorders and Stroke | NS064079 | Gavin Rumbaugh |

The funders had no role in study design, data collection and interpretation, or the decision to submit the work for publication.

## Author contributions
Thomas K Creson, Formal analysis, Investigation, Methodology, Writing—review and editing; Camilo Rojas, Murat Kilinc, Andres Jimenez-Gomez, Formal analysis, Investigation, Writing—review and editing; Ernie Hwaun, Formal analysis, Writing—review and editing; Thomas Vaissiere, Investigation, Visualization, Methodology, Writing—review and editing; Jimmy Lloyd Holder Jr, Jianrong Tang, Laura L Colgin, Courtney A Miller, Investigation, Methodology, Writing—review and editing; Gavin Rumbaugh, Conceptualization, Data curation, Formal analysis, Supervision, Funding acquisition, Investigation, Methodology, Writing—original draft

## Author ORCIDs
Gavin Rumbaugh (iD) https://orcid.org/0000-0001-6360-3894

## Ethics
Human subjects: The SYNGAP1 Patient Registry [42] (https://syngap1registry.iamrare.org) is funded through the National Organization of Rare Disorders. Collection of human subject data was reviewed and approved by Hummingbird (Study # 2016-57-SYNGAP) and Baylor College of Medicine (Study #H-30480 and #H-41411) Institutional Review Boards. The parents of patients S3-060 and S3-080, which are distinct patients from those represented in Supplementary File 3, provided written informed consent according to a protocol approved by the Baylor College of Medicine Institutional Review Board.
Animal experimentation: Animal experiments were conducted according to protocols submitted to, and approved by, Scripps Research (Protocol #15-037 and #15-038) and the Baylor College of Medicine (Protocol #AN5585) Institutional Animal Care and Use Committees.

## Decision letter and Author response
Decision letter https://doi.org/10.7554/eLife.46752.028
Author response https://doi.org/10.7554/eLife.46752.029

# Additional files

### Supplementary files
• Supplementary file 1. Pairwise comparison statistics for *Figure 1*. Pairwise comparisons for each of the four groups in data presented in *Figure 1A–B*.
DOI: https://doi.org/10.7554/eLife.46752.022

• Supplementary file 2. Pairwise comparison statistics for *Figure 1—figure supplement 1*. Pairwise comparisons for each of the four groups in data presented in *Figure 1—figure supplement 1B*.
DOI: https://doi.org/10.7554/eLife.46752.023

• Supplementary file 3. Summary of EEG data from MRD5 patients in the *SYNGAP1* Registry. Subset of entries from the *SYNGAP1* patient registry noting EEG abnormalities.
DOI: https://doi.org/10.7554/eLife.46752.024

• Supplementary file 4. Pairwise comparison statistics for *Figure 4E–F*. Pairwise comparisons for each of the four groups in data presented in *Figure 4E–F*.
DOI: https://doi.org/10.7554/eLife.46752.025

• Transparent reporting form
DOI: https://doi.org/10.7554/eLife.46752.026

### Data availability
Data used for generating figures are included in the manuscript and supporting files.

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
