## [Decision Letter]

Thank you for submitting your work entitled "Adult Reversal of Memory and Excitability Impairments in a Model of Syngap1 Pathogenicity" for consideration by *eLife*. Your article has been reviewed by three peer reviewers, and the evaluation has been overseen by Gary Westbrook as the Senior and Reviewing Editor. The following individual involved in review of your submission has agreed to reveal their identity: Mary B Kennedy (Reviewer #1). Two other reviewers remain anonymous.

As you will see from the reviewers' comments, they accepted the conclusions of the work but thought that its impact was modest at this stage given that similar observations have been made for other neurodevelopment disorders, and that the mechanistic distinction between Syngap 1 early deficits and those that were reversible in the adult are not further explored here. Thus we think the manuscript at this stage is better suited to a more specialized journal.

*Reviewer #1:*

The study by Creson et al. adds to previous work from the Rumbaugh lab on behavioral effects of synGAP haploinsufficiency. The new work shows that cognitive deficiency measured by the long-term contextual fear memory paradigm can be reversed by re-expression of a full complement of synGAP in adulthood. This finding is in contrast to their earlier finding regarding cognition measured by spontaneous alternation in a T-maze paradigm, which is believed to measure working memory. Earlier work from the Rumbaugh lab had shown that haploinsufficient mice are deficient in the alternation task, and the deficiency is not restored by re-expression of synGAP in adulthood. In that study, the authors, apparently over-generalized prematurely, by concluding that "repairing the pathogenic mutations in adulthood did not improve behavior and cognition."

In the present study, the authors further show that reduced seizure threshold produced by synGAP haploinsufficiency is reversed by re-expression in adulthood.

The study is important because it indicates that the seizure susceptibility and certain aspects of cognitive disability that accompany synGAP haploinsufficiency may be improved by correcting the defect in adulthood. The study leads to the interesting question: Which behavioral tasks are "permanently" damaged by the disruption of critical periods caused by synGAP haploinsufficiency, and which behavioral tasks are not "permanently" damaged? The answer could shed light on which brain functions can be affected by ongoing brain plasticity in adulthood.

The study appears well executed. The only major weakness of the manuscript is a lack of clarity resulting from over-generalization of results, and/or from the absence of important experimental details that should be supplied for the general readership of *eLife*.

In the Introduction, the authors again show a tendency to overgeneralize from very specific results by stating that historically, pathology leading to neurodevelopmental disorders was thought to arise exclusively from impaired brain development. They then cite several studies of particular disorders around 2008 that led to the "revolutionary idea" that "therapeutic intervention may be beneficial in adult NDD patients with fully mature brains." The authors should make the introductory paragraph more accurate, and more useful, by referring to the specific disorders (i.e. Rett Syndrome and tuberous sclerosis) for which improvement after treatment of adults was noted. It seems clear that some NDD's will not be improved by treatment in adulthood; whereas others may be. It is confusing, especially to the general readership of a journal like *eLife*, to repeatedly generalize from a few cases to all cases.

The phrase, "critical period that is essential for the assembly and excitatory balance of developing forebrain circuits" is obscure. I believe the authors mean – the " critical period during which a normal ratio of excitatory and inhibitory synapses are formed in particular brain regions. Without this normal ratio, the circuits can become overly excitable, leading to pathological function."

At the beginning of the Discussion, the authors state, "The principle finding of this study is that genetic reversal of Syngap1 pathogenicity in adult mice improves deficits in memory and excitability." This is another overgeneralization that is potentially confusing. The authors should state, "The principal finding of this study is that genetic reversal of Syngap1 pathogenicity in adult mice improves deficits in long-term contextual memory and in seizure thresholds." In the next sentence, the authors should change the ending; "… that mediate working memory." The paradox that the authors refer to arises solely from their own overgeneralization. The significance of the authors work would be more broadly accepted if they refrained from over-generalization in the first place.

*Reviewer #2:*

In this manuscript, the authors show that Syngap1 heterozygous knockout mice exhibit impaired remote memory for contextual fear when tested 30 days after training, despite having normal contextual memory when tested 1 day after training. Interestingly, the authors crossed Syngap1 Lox-stop heterozygous mice to hemizygous tamoxifen-inducible Cre mice and show that re-expressing Syngap1 in adult mice reversed the remote memory deficit. The authors proceed to demonstrate that seizure threshold, but not sensorimotor gating, phenotypes are reversed by re-expressing Syngap1 in adults. The authors conclude that genetic reversal of Syngap1 pathogenicity in adult can improve impaired cognition altered excitability.

The findings are interesting, but I am not sure whether they rise to the level of importance and novelty to warrant being published as a short report in *eLife*. As noted by the authors, similar findings in other mouse models of intellectual disability/autism spectrum disorder have demonstrated and in this manuscript there are no experiments designed to get at the mechanism or brain region where re-expression of Syngap1 is exerting this effect in adulthood. In addition, there is an interesting finding that is not explained: there is no remote memory phenotypes if the Syngap1 heterozygous mutant mice are tested one day after training, suggesting the retrieval of the memory prevents the remote memory impairment. How is does this observation impact the interpretation of the remote memory phenotype?

In closing, although the findings are interesting and potentially important for understanding pathologies in adult Syngap1 mutant mice, I am not sure that they warrant publication in *eLife*.

*Reviewer #3:*

The manuscript by Thomas Creson et al. reported that re-expression in adults can reverse some behavioral and seizure abnormalities even though SynGAP loss disrupts neurodevelopment. Specifically, the authors showed evidence that 1) re-expression of SynGAP in adults reverses remote memory deficit in a SynGap1 Het mice with 50% SynGAP loss; 2) re-expression of SynGAP in adults also alleviates seizures in SynGap1 Het mice. These findings are interesting and worth publishing but they do not reach the level of significance expected for *eLife*. The seizures and remote memory phenotypes are very interesting, but they remain mostly unexplored in this manuscript. The finding that it is possible to reverse in adults certain behavioral and physiological phenotypes in neurodevelopmental mouse mutants is interesting and clinically relevant, but it is not novel, since it has been demonstrated repeatedly in multiple neurodevelopmental conditions.

---

## [Author Response]

[Editors’ note: the manuscript was revised and accepted after re-review.]

Reviewer #1:

[…] The study appears well executed. The only major weakness of the manuscript is a lack of clarity resulting from over-generalization of results, and/or from the absence of important experimental details that should be supplied for the general readership of eLife.

*In the Introduction, the authors again show a tendency to overgeneralize from very specific results by stating that historically, pathology leading to neurodevelopmental disorders was thought to arise exclusively from impaired brain development. They then cite several studies of particular disorders around 2008 that led to the "revolutionary idea" that "therapeutic intervention may be beneficial in adult NDD patients with fully mature brains." The authors should make the introductory paragraph more accurate, and more useful, by referring to the specific disorders (i.e. Rett Syndrome and tuberous sclerosis) for which improvement after treatment of adults was noted. It seems clear that some NDD's will not be improved by treatment in adulthood; whereas others may be. It is confusing, especially to the general readership of a journal like eLife, to repeatedly generalize from a few cases to all cases.*

We have rewritten the Introduction based on the comments of the reviewer. We now draw a distinction between the earlier modeling work on syndromic risk factors that impact the whole body, and our newer work, which deals with the biology of risk factors that principally cause severe NDDs, such as SYNGAP1.

The phrase, "critical period that is essential for the assembly and excitatory balance of developing forebrain circuits" is obscure. I believe the authors mean – the " critical period during which a normal ratio of excitatory and inhibitory synapses are formed in particular brain regions. Without this normal ratio, the circuits can become overly excitable, leading to pathological function."

Thank you for pointing this out. We have changed the statement to read, “Sygnap1 heterozygosity disrupts a developmental critical period, which a normal ratio of excitatory and inhibitory synapses are formed in particular brain regions. Without this normal ratio, circuits can become overly excitable, which may contribute to pathological brain function (Clement et al., 2012; Ozkan et al., 2014).”

At the beginning of the Discussion, the authors state, "The principle finding of this study is that genetic reversal of Syngap1 pathogenicity in adult mice improves deficits in memory and excitability." This is another overgeneralization that is potentially confusing. The authors should state, "The principal finding of this study is that genetic reversal of Syngap1 pathogenicity in adult mice improves deficits in long-term contextual memory and in seizure thresholds." In the next sentence, the authors should change the ending; "… that mediate working memory." The paradox that the authors refer to arises solely from their own overgeneralization. The significance of the authors work would be more broadly accepted if they refrained from over-generalization in the first place.

We appreciate the advice from the reviewer, and we will be mindful of overgeneralizations in the future. We have rewritten the manuscript based on the substantial new data included in the study. In the new version of the manuscript, this statement has been changed to, “The principal finding in this study is that genetic reversal of Syngap1 pathogenicity in adult mice reverses deficits in remote memory, lowers seizure threshold, re-balances neural excitability, and improves hippocampal/cortical function. On the surface, these exciting results are somewhat paradoxical given that we have previously found that Syngap1 pathogenicity causes hardwired impairments in the development of forebrain circuits that mediate a subset of previously tested cognitive functions (30, 31, 47).”

Reviewer #2:

[…] The findings are interesting, but I am not sure whether they rise to the level of importance and novelty to warrant being published as a short report in eLife. As noted by the authors, similar findings in other mouse models of intellectual disability/autism spectrum disorder have demonstrated and in this manuscript there are no experiments designed to get at the mechanism or brain region where re-expression of Syngap1 is exerting this effect in adulthood.

Over the past year, we have performed substantial new experiments that: 1) get at the mechanism and brain regions that SynGAP re-expression may be exerting its influence, and 2) provide translational relevance to the mouse reversal studies. We believe that these new experiments increase the novelty and the general interest of our study.

Behavioral improvements reflecting improved cognitive function and reduced seizure susceptibility in the mouse model are now strengthened by data demonstrating a corresponding improvement in neurophysiological signals that predict seizure susceptibility and learning ability. These new studies demonstrate that protein replacement reversed both in vivo neurophysiological and behavioral phenotypes related to memory and seizure, two of the most clinically-relevant phenotypes associated with severe NDDs. Phenotypic reversal of cognitive deficits in this study was accompanied by chronic in vivo neurophysiological analysis in mice before and after gene restoration. This prospective design, which followed individual mice over two months, allowed us to measure how hippocampal and cortical function changed in individual animals after protein restoration in adulthood. In these new studies, we observed an increase in hippocampal theta power (a biomarker for memory formation) and an elimination of generalized paroxysmal spiking across the forebrain (a biomarker for seizure susceptibility) in mutant mice that underwent gene restoration. These clear improvements in hippocampal and cortical function provide mechanistic insight into how behavioral measures of seizure and memory improved after protein re-expression in adult animals. The combined improvement in both in vivo neurophysiology and behavior in response to adult gene repair provides very strong evidence that increasing the levels of pathologically low protein caused by severe genetic variants can broadly improve the function of the hippocampus and cortex even in mature animals.

We also now provide data from human SYNGAP1 patients that provide translational context to the electrophysiological studies in mice that are described above. We collected data from human patients that harbor similar genetic pathogenicity as the mouse models (i.e. haploinsufficiency of SYNGAP1/Syngap1). We found that both mouse models and human patients exhibited analogous pathological EEG waveforms, which was the tendency of interictal activity to increase during periods of sleep. In mice, the pathological waveforms biased toward sleep were abolished after adult-initiated protein restoration. Therefore, these pathological EEG waveforms are a promising candidate biomarker for generalized cortical dysfunction in this genetically-defined NDD. We have confidence in the validity of these signals as a candidate biomarker because the signals worsened during sleep in both species. Sleep-driven worsening of pathological EEG signals is associated with declining cognitive function in NDD patients. Therefore, abolishment of these signals in mice, combined with improved behavioral measures of long-term memory, suggest that EEG signals can be a proxy measure for generalized improvements in cognitive function in single-gene causes of NDDs. Past clinical trials for NDD treatments searching for improved cognitive function in patients have relied on subjective endpoints, such as parent-reported surveys. These endpoints suffer from a very high placebo effect, which can run up to 30%. EEG studies are easily performed in the clinic, even in children with severe NDDs, and signals collected from EEG are highly quantifiable. The objective nature of EEG signals suggest that they may be less prone to measurement bias leading to a lower placebo effect, and therefore may be more useful in clinical trials that test the efficacy of emerging therapies for genetically-defined NDDs.

In addition, there is an interesting finding that is not explained: there is no remote memory phenotypes if the Syngap1 heterozygous mutant mice are tested one day after training, suggesting the retrieval of the memory prevents the remote memory impairment. How is does this observation impact the interpretation of the remote memory phenotype?

We reasoned that if retrieval at 1-day after training can prevent 30-day retrieval deficits, then networks that either consolidate and/or retrieved remote memory must be structurally intact in adult Syngap1 mice. We therefore hypothesized that these structurally-intact memory-linked circuits likely suffer from functional deficits caused by the persistent reduction of SynGAP protein expressed in the forebrain. It follows that restoring SynGAP protein in these circuits, even in adulthood, would improve memory performance in these mice. Our behavioral data supports this hypothesis. Importantly, our new in vivo ePhys data also supports this conclusion because we see clear evidence of improved hippocampal and cortical function in Syngap1 mutants after protein re-expression in adulthood. Moreover, we have shown in a past study, that adult restoration of SynGAP protein in Syngap1 Hets also rescues LTP in the CA1. Together, these data support the hypothesis that the Syngap1 gene retains functions in the adult brain to control the function and plasticity of forebrain circuits that promote memory and balance network excitability.

In closing, although the findings are interesting and potentially important for understanding pathologies in adult Syngap1 mutant mice, I am not sure that they warrant publication in eLife.

We hope that the addition of human EEG data and the in vivo ePhys studies in Syngap1 rescue mice raise the enthusiasm of the reviewer.

Reviewer #3:

The manuscript by Thomas Creson et al. reported that re-expression in adults can reverse some behavioral and seizure abnormalities even though SynGAP loss disrupts neurodevelopment. Specifically, the authors showed evidence that 1) re-expression of SynGAP in adults reverses remote memory deficit in a SynGap1 Het mice with 50% SynGAP loss; 2) re-expression of SynGAP in adults also alleviates seizures in SynGap1 Het mice. These findings are interesting and worth publishing but they do not reach the level of significance expected for eLife. The seizures and remote memory phenotypes are very interesting, but they remain mostly unexplored in this manuscript. The finding that it is possible to reverse in adults certain behavioral and physiological phenotypes in neurodevelopmental mouse mutants is interesting and clinically relevant, but it is not novel, since it has been demonstrated repeatedly in multiple neurodevelopmental conditions.

We thank the reviewer for the insightful comments. The major concern, that the reversal of behaviors in adulthood in Syngap1 mice was largely unexplored in the original manuscript, has been dealt with in the revised manuscript. We now include two new and substantial experiments in the revised study to further explore the reversal phenotypes in this mouse model. We believe that these new experiments increase the novelty and the general interest of our study.

Behavioral improvements reflecting improved cognitive function and reduced seizure susceptibility in the mouse model are now strengthened by data demonstrating a corresponding improvement in neurophysiological signals that predict seizure susceptibility and learning ability. These new studies demonstrate that protein replacement reversed both in vivo neurophysiological and behavioral phenotypes related to memory and seizure, two of the most clinically-relevant phenotypes associated with severe NDDs. Phenotypic reversal of cognitive deficits in this study was accompanied by chronic in vivo neurophysiological analysis in mice before and after gene restoration. This prospective design, which followed individual mice over two months, allowed us to measure how hippocampal and cortical function changed in individual animals after protein restoration in adulthood. In these new studies, we observed an increase in hippocampal theta power (a biomarker for memory formation) and an elimination of generalized paroxysmal spiking across the forebrain (a biomarker for seizure susceptibility) in mutant mice that underwent gene restoration. These clear improvements in hippocampal and cortical function provide mechanistic insight into how behavioral measures of seizure and memory improved after protein re-expression in adult animals. The combined improvement in both in vivo neurophysiology and behavior in response to adult gene repair provides very strong evidence that increasing the levels of pathologically low protein caused by severe genetic variants can broadly improve the function of the hippocampus and cortex even in mature animals.

We also now provide data from human SYNGAP1 patients that provide translational context to the electrophysiological studies in mice that are described above. We collected data from human patients that harbor similar genetic pathogenicity as the mouse models (i.e. haploinsufficiency of SYNGAP1/Syngap1). We found that both mouse models and human patients exhibited analogous pathological EEG waveforms, which was the tendency of interictal activity to increase during periods of sleep. In mice, the pathological waveforms biased toward sleep were abolished after adult-initiated protein restoration. Therefore, these pathological EEG waveforms are a promising candidate biomarker for generalized cortical dysfunction in this genetically-defined NDD. We have confidence in the validity of these signals as a candidate biomarker because the signals worsened during sleep in both species. Sleep-driven worsening of pathological EEG signals is associated with declining cognitive function in NDD patients. Therefore, abolishment of these signals in mice, combined with improved behavioral measures of long-term memory, suggest that EEG signals can be a proxy measure for generalized improvements in cognitive function in single-gene causes of NDDs. Past clinical trials for NDD treatments searching for improved cognitive function in patients have relied on subjective endpoints, such as parent-reported surveys. These endpoints suffer from a very high placebo effect, which can run up to 30%. EEG studies are easily performed in the clinic, even in children with severe NDDs, and signals collected from EEG are highly quantifiable. The objective nature of EEG signals suggest that they may be less prone to measurement bias leading to a lower placebo effect, and therefore may be more useful in clinical trials that test the efficacy of emerging therapies for genetically-defined NDDs.